# Scale-invariant large nonlocality in polycrystalline graphene

Mário Ribeiro[1,6], Stephen R. Power [2,3], Stephan Roche[2,4], Luis E. Hueso [1,5] & Fèlix Casanova [1,5]

The observation of large nonlocal resistances near the Dirac point in graphene has been related to a variety of intrinsic Hall effects, where the spin or valley degrees of freedom are controlled by symmetry breaking mechanisms. Engineering strong spin or valley Hall signals on scalable graphene devices could stimulate further practical developments of spin- and valleytronics. Here we report on scale-invariant nonlocal transport in large-scale chemical vapor deposition graphene under an applied external magnetic field. Contrary to previously reported Zeeman spin Hall effect, our results are explained by field-induced spin-filtered edge states whose sensitivity to grain boundaries manifests in the nonlocal resistance. This phenomenon, related to the emergence of the quantum Hall regime, persists up to the millimeter scale, showing that polycrystalline morphology can be imprinted in nonlocal transport. This suggests that topological Hall effects in large-scale graphene materials are highly sensitive to the underlying structural morphology, limiting practical realizations.

[1] CIC nanoGUNE, 20018 Donostia-San Sebastian, Basque Country, Spain. [2] Catalan Institute of Nanoscience and Nanotechnology (ICN2), CSIC and The Barcelona Institute of Science and Technology, Campus UAB, 08193 Bellaterra, Catalonia, Spain. [3] Universitat Autònoma de Barcelona, 08193 Bellaterra, Catalonia, Spain. [4] ICREA—Institució Catalana de Recerca i Estudis Avançats, 08010 Barcelona, Catalonia, Spain. [5] IKERBASQUE, Basque Foundation for Science, 48013 Bilbao, Basque Country, Spain. [6] Present address: Center for Quantum Nanoscience, Institute for Basic Science (IBS), Seoul 03760, Republic of Korea. Correspondence and requests for materials should be addressed to L.E.H. (email: l.hueso@nanogune.eu) or to F.C. (email: f.casanova@nanogune.eu)

In recent years, there has been a continuous effort to harvest the spin and valley degrees of freedom of charge carriers for developing innovative information processing as an alternative to complementary metal-oxide-semiconductor technologies[1–7]. The high-mobility, low spin–orbit coupling (SOC), and linear energy dispersion of graphene has made it a core platform in such quest. In this context, graphene has been explored in magnetic-element-free nonlocal transport using Hall-bar geometries. The large nonlocal signals are observed close to the Dirac point, and have been tentatively related to topological Hall effects such as the spin Hall effect (SHE)[2,8–10], or long-range chargeless topological valley Hall currents for gapped graphene/h-BN hetero-structures[11,12]. Spin signals in graphene using nonlocal approaches free of magnetic elements have been reported by applying external magnetic fields[3,13], by using extrinsic sources of SOC[2,8–10,14], and by proximity effect to ferromagnetic insulators[15]. Theoretical predictions have mainly explored spin diffusion in the spin Hall regime, where spin currents are respectively generated and detected by means of the SHE and the inverse spin Hall effect (ISHE)[9,16–18]. Nonlocal signatures related to valley Hall currents have also been predicted to emerge from the different diffraction features of the two valleys in graphene[19]. The theoretical frameworks in the spin Hall regime consider extrinsic sources of SOC, Zeeman interaction (Zeeman spin Hall effect), and proximity-induced strong-exchange bias as possible mechanisms driving such enhancements of nonlocal signals close to the Dirac point[18,20–23]. More recently, spin generation and detection via extrinsic sources of SOC have been questioned by Wang et al.[24], Kaverzin et al.[25], and Cresti et al.[26]

All current demonstrations however rely on microscale Hall bars of pristine graphene obtained from micromechanical exfoliation techniques, with natural flakes with dimensions ranging several micrometers, or on microscale Hall bars patterned from chemical vapor deposition (CVD) graphene[2,3,8,10,13,15]. The micrometer spin and valley relaxation lengths therefore restrict the experimental realizations to such scale. Core in the analysis of nonlocal signals, these approaches rely on comparing the nonlocal signal to the background arising from the classical current spreading (Ohmic) contribution and studying its dependence with the channel geometry to sustain the claim of other-than-charge sources for the nonlocality.

Here, we demonstrate the persistence of large nonlocal signals at the Dirac point of CVD graphene devices from the micrometer up to the millimeter scale in presence of an external magnetic field applied perpendicular to the surface. These signals exhibit a similar dependence with the magnetic field to those induced via Zeeman spin Hall effect[3] in microscale high-quality pristine graphene devices, and closely follow the same dependence with the device aspect ratio for device lengths that differ by two orders in magnitude. We however exclude an origin based on pure spin and valley currents due to the length scales involved in the transporting channel of the macroscale devices, and conclude that the origin of the intriguing nonlocality for both the microscale and macroscale stems from the same fundamental mechanism. Considering the microscopic details of the fabricated samples and the polycrystalline morphology of the CVD graphene samples, the large nonlocal signals are consistent with a dissipative quantum Hall regime driven by Zeeman-split counter-propagating edge states[16,20,27,28]. Additionally, the grain boundaries (GBs) shunt the insulating bulk[28–33] induced by the strong magnetic field, and generate a strong asymmetry of the nonlocal magneto-resistance with external magnetic field, a striking feature already observed, but not yet understood[3].

## Results

**Device fabrication and measurement configuration.** The availability of mm-sized continuous CVD graphene films enables the fabrication of Hall bars in a wide range of scales, and the study of nonlocal signals for device geometries not achievable using standard micromechanical exfoliation techniques[34]. The devices studied in this work were fabricated following standard electron-beam lithography procedures on monolayer CVD graphene films wet-transferred onto $1 \times 1 \, cm^2$ Si/SiO$_2$(300 nm) substrates (see Methods). Standard local electrical characterization was first employed to ensure the quality of the metal contacts, and that the graphene samples exhibited characteristic transport properties unique to graphene, such as the half-integer quantum Hall effect[35] (see Supplementary Notes 1 and 2). Figure 1a shows a sketch of a Hall bar depicting the measurement configuration of the nonlocal transport, and the two sources of nonlocal signals mainly discussed in this paper. Figure 1b shows an optical microscope picture of the macroscale sample.

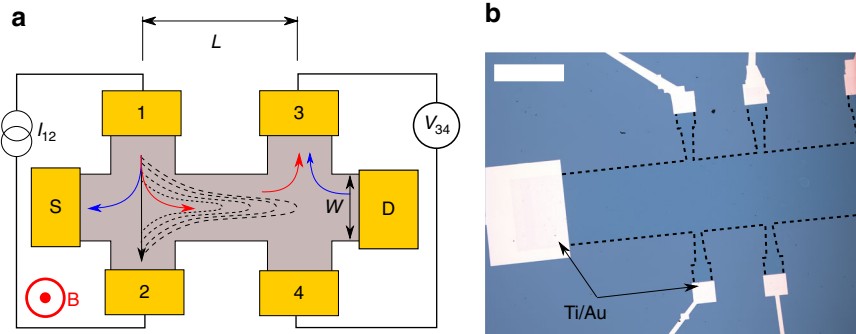

**Fig. 1** Device structure and nonlocal measurements. **a** Sketch of the Hall-bar-shaped channel for nonlocal measurements. The nonlocal measurement consists of driving a current (black arrow) across contacts 1 and 2, and measuring the resulting voltage signal between probes 3 and 4. The nonlocal resistance is defined as $R_{NL} = V_{34}/I_{12}$. In the spin Hall regime, a spin current transverse to the injected charge current is generated via SHE (spin "down", blue arrow; spin "up", red arrow), and diffuses through the channel of length $L$ until it reaches electrodes 3 and 4, where via ISHE the spin current generates a charge current. The black dashed lines represent the van der Pauw contribution to the nonlocal resistance detected between probes 3 and 4. This background signal originates from the current spreading from terminals 1 and 2, and depends only on the resistivity of the conducting channel multiplied by an exponential decaying geometrical factor. In our devices, the current is always injected in the left arm, and detected at the right arm, with the external magnetic field applied perpendicular to the surface. **b** Optical microscope image of the macroscale CVD graphene sample. The channel width, $W$, is 500 μm, with the electrodes distance $L$ being 500, 700, 900, 1200, and 1500 μm. The width of the contacts is defined to be 1/10th of the width of the channel. White scale bar is 500 μm

In a concise manner, nonlocal measurements in the context of spin Hall currents consist of driving a current in one arm of an H-bar shaped channel and detecting the voltage drop at the other arm (see Fig. 1a). Using the terminal notation shown in Fig. 1a, the nonlocal resistance is determined as $R_{NL} = V_{34}/I_{12}$. If the channel length matches the spin diffusion length in the non-magnetic material, this simple device scheme and measurement setup makes possible the detection of spin-related signals in the spin Hall regime. The Hall-bar geometry allows for the generation of a transverse spin current from the charge current via SHE, which will then diffuse across the channel and be converted back into a charge current via ISHE. The magnitude of the effect will depend on the efficiency of the spin-to-charge conversion, and on the spin relaxation length. These two quantities can be determined from a transmission line method measurement of the nonlocal resistance[16], where the nonlocal resistance is related to the length, $L$, and width, $W$, of the transporting channel as $R_{NL} = \frac{1}{2}\theta_{SH}^2 \rho_{xx} \frac{W}{\lambda_s} \exp\left(-\frac{L}{\lambda_s}\right)$, where $\theta_{SH}$ is the spin Hall angle, $\rho_{xx}$ the sheet resistance, and $\lambda_s$ the spin diffusion length. In graphene, $R_{NL}$ in the spin Hall regime is greatly enhanced at the Dirac point, requiring the use of a gate voltage to sweep the Fermi level of graphene to the charge neutrality point (CNP)[2,3,8,10,13,15].

Although nonlocal measurements are used to probe non-charge-related transport, there are classical, charge-related sources of nonlocality that can contribute to the signal detected between terminals 3 and 4 (see Fig. 1a). A robust source of nonlocal signals is the classical van der Pauw current spreading from the injecting terminals[3,36,37]. By injecting a current on the left arm, a net current will reach the detection terminals, with magnitude decreasing exponentially with the distance to the injecting terminals. This Ohmic contribution in the device scheme presented is determined using the formula $R_{NL,Ohmic} = \frac{\rho_{xx}}{\pi}\ln\left[\frac{\cosh(\pi L/W)+1}{\cosh(\pi L/W)-1}\right]$, which, for cases where $L > W$, is usually approximated as $R_{NL,Ohmic} \approx \frac{4}{\pi}\rho_{xx}\exp\left(-\pi\frac{L}{W}\right)$. The sheet resistance is determined by performing a four-probe measurement of the respective channel, injecting a current between electrodes S and D, and measuring the voltage drop between electrodes 2 and 4, or 1 and 3. By comparing the detected nonlocal resistance vs. the expected nonlocal Ohmic contribution, we can evaluate the emergence of signals not related to this classical source.

In our study, the nonlocal resistance at the CNP of graphene is measured for different applied magnetic fields ($B$), and then compared to the Ohmic contribution evaluated under the same conditions. Throughout the manuscript, we keep the same relative orientation of the perpendicular $B$ and of the arms injecting the current and detecting the voltage signal. We explored CVD graphene Hall bars with channel 5, 50, and 500 μm wide, with center-to-center distance between terminals maintaining similar aspect ratios ($L/W$) of typically 1, 1.4, 1.8, 2.4, and 3.2. The width of the terminals is 1/10th or less of the channel width (see Methods). We report mainly on the results obtained for the extreme cases of 5 and 500-μm-wide samples, using the data of the 50-μm-wide sample to extend the discussion. Further details on the fabrication and electrical measurements are provided in Methods.

**Nonlocality in macroscale devices**. Figure 2 shows the results obtained for the 500-μm-wide macroscale CVD graphene sample, for a channel length of 1500 μm, at cryogenic temperatures.

In Fig. 2a, a sweep of the gate voltage, $V_G$, in the absence of magnetic field reveals the carrier density dependence of graphene, with CNP at $V_G = 34$ V. The Ohmic background closely follows

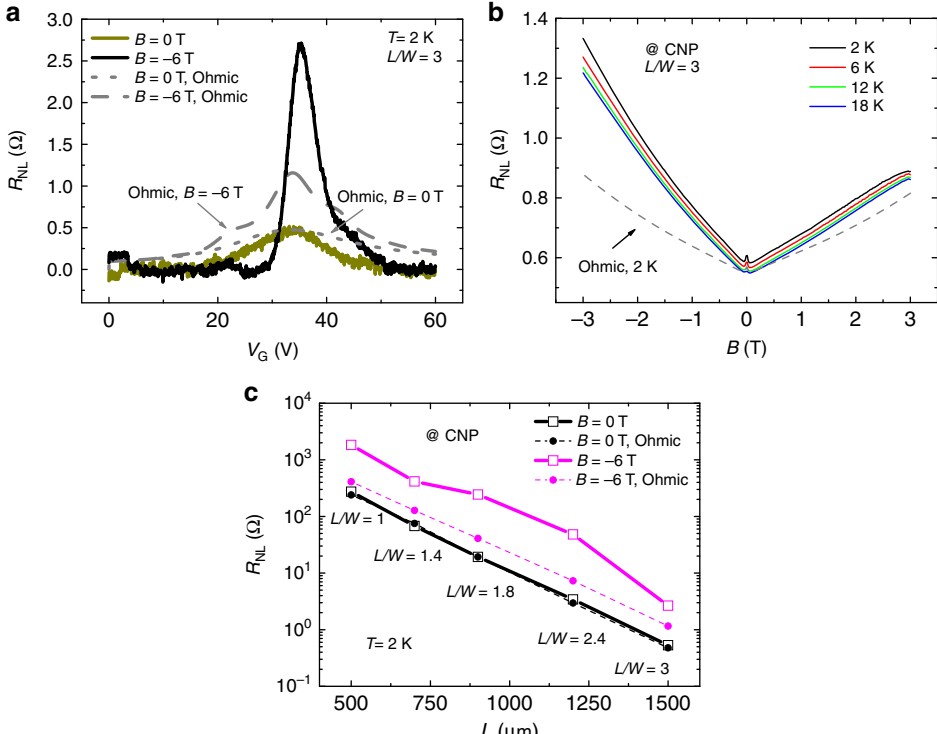

**Fig. 2** Nonlocality in macroscale CVD graphene devices at low temperatures. **a** Nonlocal resistance as a function of back-gate voltage for a 500-μm-wide and 1500-μm-long Hall bar. **b** Nonlocal resistance as a function of the applied out-of-plane magnetic field, at the CNP of graphene, for different temperatures. **c** Transmission line plot of the nonlocal resistance with channel length, at the CNP. In all figures, dashed lines represent the determined Ohmic contributions from four-probe measurements under equivalent conditions

the nonlocal resistance and, at the CNP, matches the detected nonlocal resistance. When $B$ is applied, a significant increase of $R_{NL}$ at the Dirac point occurs, together with a narrowing of the curve (see Supplementary Note 3 for the temperature dependence). This effect is not reproduced by the Ohmic contribution. By fixing the gate voltage at the CNP and sweeping $B$ (Fig. 2b), one verifies two fundamental features. Firstly, the nonlocal magnetoresistance is asymmetric, with significantly higher nonlocal values for negative values of $B$. Secondly, for the side with higher signals, the difference between the nonlocal resistance and the Ohmic contribution increases with increasing $B$. Furthermore, the Ohmic signal shows a more symmetric magnetoresistance.

Expanding the study to the different channel lengths and focusing on the side with highest nonlocal signal, Fig. 2c shows the dependence of $R_{NL}$ and of the Ohmic contribution at 2 K as a function of the channel length, at the CNP, with ($B = -6$ T) and without ($B = 0$) applied magnetic field. Without magnetic field, the expected Ohmic contribution and the measured $R_{NL}$ coincide, following the same dependence with the channel length. Fitting the dependence to an exponential decay, the resulting decay $\lambda = 159.4 \pm 1.6\,\mu$m closely matches the expected $W/\pi$ from the Ohmic term $\exp\left(-\pi \frac{L}{W}\right)$, calculated to be ~159.2 μm. Upon switching on the magnetic field, $R_{NL}$ follows a similar exponential decay with channel length, but the magnitude of the resistance is greatly enhanced, in some cases being one order of magnitude higher than the Ohmic signal for similar conditions. If the signal were to be interpreted in terms of charge neutral spin currents, (or, equivalently, valley currents), a fitting to the expression $R_{NL} = \frac{1}{2}\theta_{SH}^2 \rho_{xx} \frac{W}{\lambda_s} \exp\left(-\frac{L}{\lambda_s}\right)$ would yield a spin relaxation length of $\lambda_s = 163 \pm 19\,\mu$m and $\theta_{SH} = 1.6 \pm 0.4$. Such values are clearly unreasonably high for such a disordered polycrystalline graphene sample

supported onto an oxide substrate. Actually, the spin Hall angle determined is one order of magnitude higher than what is typical of metals with strong SOC, such as platinum ($\theta_{SH} \sim 0.2$)[5,38].

**Nonlocality in microscale devices.** Figure 3a shows the nonlocal resistance measured for the 5-μm-wide microscale CVD graphene sample, for a channel length of 16 μm, at cryogenic temperatures.

Similarly to what was found before, in the absence of an applied magnetic field ($B = 0$), and by sweeping the gate voltage, $R_{NL}$ is well described by the Ohmic background, with a decay $\lambda = 1.58 \pm 0.03\,\mu$m. However, in this case, under an applied magnetic field ($B = 6$ T), the enhancement of the nonlocal signal is one order of magnitude higher than in the macroscale sample for an equivalent aspect ratio, $L/W = 3$, with $R_{NL}$ two orders of magnitude higher than the Ohmic background (see Supplementary Note 3 for the temperature dependence). Strikingly, fixing the gate voltage at the CNP and sweeping $B$ (Fig. 3b), the magnetoresistance of the graphene becomes greatly asymmetric, but this time the higher nonlocal values happen for positive values of $B$. A strong asymmetry of the nonlocal resistance with $B$ was also reported in other studies of nonlocality, but such effects were not analyzed[3]. Evaluating $R_{NL}$ at the CNP for the different channel lengths at 2 K with ($B = 6$ T) and without ($B = 0$) applied magnetic field, Fig. 3c exhibits a similar trend to that reported for the macroscale sample, with a clear agreement between the nonlocal resistance and the Ohmic contribution at $B = 0$, following the same dependence with the channel length. In the presence of $B$, the nonlocal resistance dependence is also clearly described by a similar exponential decay with channel length, but the magnitude of the signals is even further enhanced, from one to two orders of magnitude larger than the expected Ohmic background. An analysis in terms of non-charge currents yields

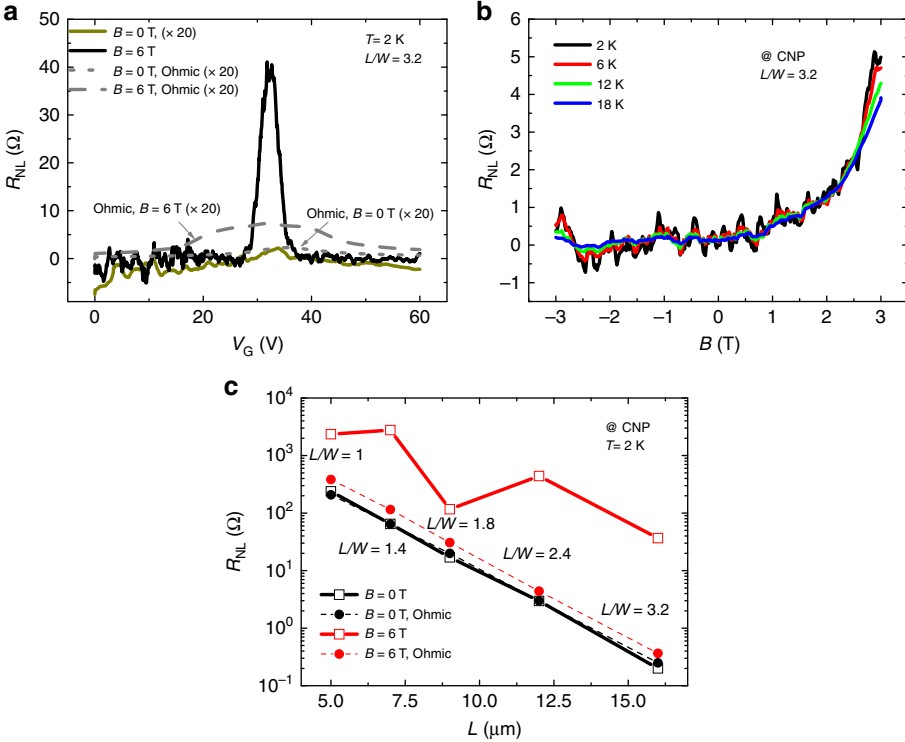

**Fig. 3** Nonlocality in microscale CVD graphene devices at low temperatures. **a** Nonlocal resistance as a function of back-gate voltage for a 5-μm-wide and 16-μm-long Hall bar. **b** Nonlocal resistance as a function of the applied out-of-plane magnetic field, at the CNP of graphene, for different temperatures. **c** Transmission line plot of the nonlocal resistance with channel length, at the CNP. In all figures, dashed lines represent the determined Ohmic contributions from four-probe measurements under equivalent conditions

$\lambda_s = 2.7 \pm 1.2\ \mu m$, and $\theta_{SH} = 1.3 \pm 0.6$. We note that this relaxation length is similar to reports that place $\lambda_s$ between 1 and 5 μm in CVD graphene on SiO$_2$[39].

## Discussion

So far, we have demonstrated that CVD graphene samples that differ in dimensions by two orders of magnitude can show similar large nonlocal signals close to the Dirac point, which dominate over the conventional Ohmic contribution. In the absence of magnetic field, these samples exhibit nonlocal resistance in perfect agreement with what is expected from the van der Pauw background. The source of the strong nonlocal signal when the magnetic field is applied is less clear and more complex. In the macroscale devices, an origin related to spin diffusion within the spin Hall regime would require the spins generated in the injecting terminal to survive a disordered 1.6-mm-long channel, and convert back into a charge current, which disagrees with all the estimations of spin diffusion lengths reported to date in CVD graphene samples[39]. This strongly suggests that the signal visible in the macroscale sample is not related to a pure spin transport mechanism within an SHE and ISHE process. At the microscale sample, the signal increases even further, by one order of magnitude. Again, the dependence with channel length indicates that the signal decays exponentially similarly to that predicted from the Ohmic contribution of 1.59 μm.

Besides the origins related to spin transport and van der Pauw backgrounds, thermoelectric effects have also been proposed as a possible mechanism driving nonlocality at the CNP of graphene, in particular the Ettingshausen–Nernst effect[13,40]. In this picture, the nonlocal signal would be generated in two steps: first, under an applied perpendicular magnetic field, the current being driven would lead to a transverse heat flow (Ettingshausen effect); second, the thermal gradient across the detection terminal under $B$ would generate the nonlocal voltage (Nernst effect). Since thermoelectric coefficients in graphene show a strong increase around the Dirac point, the gate dependence should manifest an enhancement at the CNP. But considering both Ettingshausen and Nernst effects, the dependence of $R_{NL}$ with $B$ should be quadratic, $R_{NL} \propto B^2$. This seems not to be the case in our samples, neither in the microscale sample (Fig. 3b) nor in the macroscale one (Fig. 2b), where the dependence of the nonlocal resistance with magnetic field is strongly asymmetric.

From this discussion, the nonlocal signals in our experiment do not seem to originate from spin or valley Hall effects, thermoelectric effects, or purely Ohmic contributions. To further clarify the nature of the observed features, we repeated the experiments for a sample with dimensions in between 5 and 500 μm. With a 50-μm-wide channel, and equivalent aspect ratios between 1 and 3.2, the sample is on the macroscale side, and should exclude spin transport. Figure 4a summarizes the nonlocal resistance at the CNP of graphene taken at $|B| = 6\ T$ as a function of the channel aspect ratio, for all three samples (5-, 50-, and 500-μm-wide channel).

The nonlocal resistance for the 50-μm-wide sample remains between those obtained for the two other samples, with signals 1−2 orders of magnitude higher than expected from the van der Pauw background. Interestingly, the signal magnitude is similar to the 5-μm-wide sample, where spin signals could conceivably survive a 16-μm-long channel. In this case, with a 160-μm-long channel, we exclude a spin transport origin. The mechanism driving the nonlocal response must be able to generate signals dependent with the magnitude and sign of the external magnetic field, seemingly invariant with scale, and that follow a dependence with channel length similar to the Ohmic background. To explain all these anomalous features, we propose an interpretation based on counter-propagating edge states shunted by GBs.

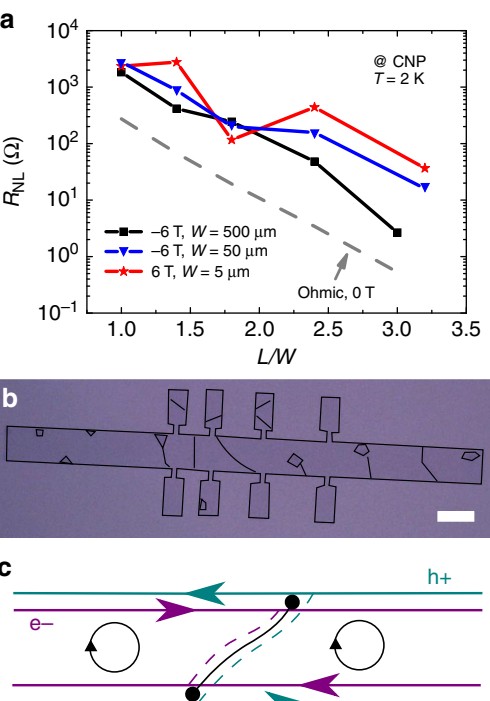

**Fig. 4** Origin of nonlocal resistance for the CVD graphene samples. **a** Nonlocal resistance as a function of aspect ratio of the Hall bars for the 5-, 50-, and 500-μm-wide samples, at CNP and 2 K. **b** Optical image of the microscale CVD graphene sample during the fabrication marking the visible grain boundaries and bilayer islands. White scale bar is 5 μm. **c** At the CNP, and under an external field, the counter-propagating edge currents are coupled to the charge of the carrier via Zeeman interaction. The presence of hole/electron asymmetric grain boundaries shunting the edge states through the insulating bulk leads to nonlocal measurements that are sensitive to the sample topology and sign of the magnetic field

When an out-of-plane magnetic field is applied to graphene, the bulk gradually becomes insulating, while charge current increasingly flows at the edges, resulting in a well-defined quantum Hall effect for large fields (see Supplementary Note 2). The uniqueness of the electronic properties of graphene give rise to a zeroth Landau level (LL0) at the CNP which is populated by both electrons and holes[34,35,41]. The Zeeman splitting of the LL0 couples the charge carrier type to the spin state, so that electrons and holes are now associated with opposite spin polarizations, which triggers a dissipative quantum Hall effect near the CNP, driven by spin-polarized counter-propagating edge states (as shown in Fig. 4c)[28,42]. A key feature of this effect is that the longitudinal resistivity $\rho_{xx}$ at the CNP is now predominantly driven by edge, and not bulk, channels[28]. Under these conditions, and considering a four-point contact configuration similar to that explored in our work, a nonlocal signal emerges at the CNP solely driven by potential drops at the contacts (see Supplementary Note 4 for a multi-terminal Landauer–Büttiker formulation of the device detailing the origin of the nonlocality). A similar fingerprint (peak in $R_{NL}$ at the CNP) has already been found in a nonlocal measurement on two-dimensional system with a simultaneous presence of electrons and holes in a 20 nm HgTe quantum well[43].

In our experimental data, the presence of a strong asymmetric behavior of $R_{NL}$ with the direction of $B$ is a salient observation that demands interpretation. The strong nonlocal signal can be almost entirely quenched by changing the sign of $B$. The quenching is consistent with bulk conduction, meaning that a

shunting of the edge currents that drive the nonlocal signal enter into play. Indeed, previous studies associate conducting bulk states with a decay of the $\rho_{xx}$ peak away from the CNP[28]. In CVD graphene, line defects (i.e., GBs) are the most likely source of this conduction through the bulk, and as shown in Fig. 4b, our nonlocal transport geometry intercepts many GBs along the transport channel. Importantly, such defects can display strong electron–hole asymmetry in their transmission properties due to local sublattice symmetry breaking[29–33,44,45] or by intercepting electron–hole puddles across the device. This manifests as an asymmetry with respect to magnetic field direction due to the Zeeman splitting, which associates each charge carrier type with a different spin orientation. In Supplementary Note 4, we explicitly demonstrate how nonlocal resistances develop an asymmetry with respect to the sign of an applied magnetic field in the presence of GBs. It is important to note that in the context of quantum Hall experiments a strong dependence of $\rho_{xx}$ with the sign of $B$[46–52] or a signature of edge transport in nonlocal magnetotransport measurements[53–55] does not necessarily require a graphene-specific model. In particular, an asymmetry of $\rho_{xx}$ with $B$ has been observed in well-behaved semiconductor two-dimensional electron gases, with mechanisms such as conduction paths in the bulk[47], carrier density gradients[48,49], hybrid constrictions[50], and in-plane electric fields[51,52] driving the effect. Our proposed mechanism, however, differs in that it can explain not only the asymmetry of the nonlocal resistance but also the large nonlocal signals beyond the Ohmic contribution at the CNP. The physics of the LL0 in graphene, with a coexistence of electrons and holes (and thus a sensitivity to electron–hole asymmetries), has no equivalent in other low-dimensional systems. To demonstrate that GBs are at the origin of the $B$ asymmetry of the nonlocal signal, we perform a tight-binding simulation in the nonlocal geometry with a single 558-type line defect connecting the edges (see Methods) of a 40-nm-wide and 85-nm-long graphene channel. As seen in Fig. 5a, a highly asymmetric nonlocal peak at the CNP emerges when a Zeeman splitting is introduced in the presence of a line defect. This follows directly from the electron–hole asymmetry of transmission through the defect, discussed in further detail in Supplementary Notes 4 and 5. The similarity between the modeled system and the experimental results shown in Fig. 5b for the device with $L/W = 3.2$ of the 5-μm-wide sample strongly suggests counter-propagating edge states shunted by GBs to be at the root of the large, asymmetric nonlocality observed in the samples. An extended study over all devices with different aspect ratios of the 5-μm-wide sample shows a strong asymmetry of the nonlocality

with the sign of the magnetic field for all devices, further corroborating our analysis (see Supplementary Note 6). While all the previous discussed origins could not address this strong asymmetry, our proposed mechanism explains it. Furthermore, our simulations capture additional peaks associated with the first Landau level (LL1). A finite $R_{NL}$ is here associated with conduction through bulk states and not to counter-propagating states, which only occur near the CNP. Higher LL peaks in $R_{NL}$ follow $\rho_{xx}$, a feature previously demonstrated in other works[3,11].

Due to the random formation of inhomogeneities and GBs during the growth of CVD graphene, samples with individual, random line defects can show nonlocality preferentially for either positive or negative magnetic fields. Electron–hole asymmetry in GB transmissions can arise due to both intrinsic sublattice effects and external doping effects. Other considerations, including the positioning of GBs relative to the probes, the rate of backscattering between counter-propagating states, and the density of electron–hole puddles, can play an additional role in determining the exact nonlocal signature. However, the qualitative effect (a nonlocal peak at the CNP whose bulk-mediated suppression is dependent on the sign of $B$) is very general and, in principle, independent of sample size. However, larger samples contain more grains across their width and the asymmetric effects of individual GBs will tend to be averaged out. This will lead to smaller asymmetries with the direction of $B$, but also smaller nonlocal signals due to a greater number of bulk conduction channels. This picture fits convincingly with the range of effects exhibited by our measured graphene samples from the microscale to the macroscale. Importantly, the appearance of a strong nonlocal signal is independent of spin or valley transport mechanisms arising from, for example, long-ranged coherent spin transport and the SHE/ISHE effects.

In summary, we identify strong nonlocal signals in both microscale and macroscale CVD graphene Hall-bar devices, emerging when a perpendicular magnetic field is applied. The observed signals share many similarities to experiments relying on spin Hall or valley Hall effect mechanisms, but our control experiments at the macroscale rule out both long-range spin and valley polarized transport. The similarity of the nonlocal phenomenon across different scales suggests that a different mechanism is at the origin of such nonlocality. We propose a mechanism driven by field-induced spin-split edge states. The sensitivity of these states to electron–hole asymmetric transport, induced by features such as GBs, strongly influences the nonlocal resistance profiles. The persistence of this behavior to millimeter length scales shows that defect-induced contributions to nonlocal

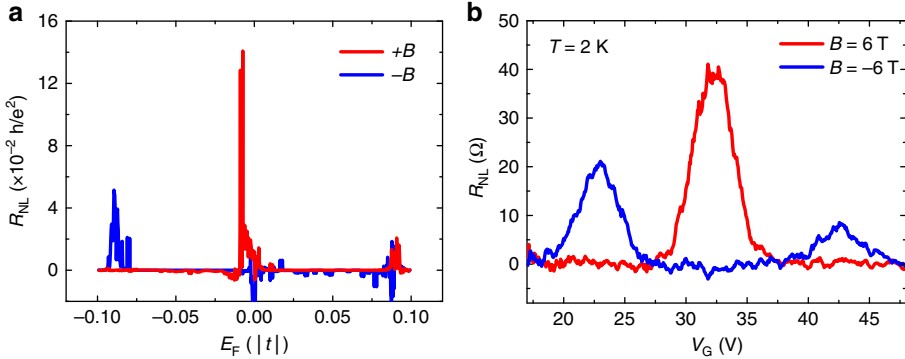

**Fig. 5** Nonlocal resistance dependence with magnetic field direction in a graphene Hall bar device. **a** Tight-binding simulation in the nonlocal geometry with a single 558-type line defect connecting the edges of a Hall-bar graphene channel. Nonlocal signals vs. position of the Fermi level relative to the CNP for opposite directions of the applied perpendicular magnetic field. **b** Nonlocal resistance as a function of back-gate voltage for a 5-μm-wide and 16-μm-long graphene Hall bar at 2 K for opposite directions of the applied perpendicular magnetic field

transport could emerge in a wide range of measurements, and may indeed dominate over more exotic sources of nonlocality in practical devices made from scalable graphene materials.

## Methods

**Device fabrication.** The same fabrication procedures were followed for the fabrication of the 5-, 50-, and 500-μm-wide CVD graphene samples. 1×1 cm$^2$ Si(n++ doped)/SiO$_2$(300 nm) chips with monolayer CVD graphene grown on copper foils were acquired from a commercially available supplier, Graphenea S.A. All samples reported here come from the same batch. The samples were first spin coated with double-layer of Poly(methyl methacrylate) (PMMA) (495/950 kDa) and then baked at 180 °C for 90 s. To define the Hall-bar-shaped graphene, the areas surrounding the Hall bar were exposed with electron-beam lithography, and etched with a chemistry of Ar/O$_2$ (80/5 sccm) in a capacitive coupled plasma reactive-ion etching Oxford PlasmaLab 80 equipment. The remaining resist leftovers were striped in acetone bath at room temperature for 4 h, immersed in isopropanol, and dried with a nitrogen gun. For the definition of the electrical contacts, the samples were again spin coated with double-layer PMMA, and the metal electrodes defined over the Hall bar. The metallization was done on ultra-high vacuum at a base pressure of 10$^{-9}$ mbar, using electron-beam deposition of Ti (5 nm)/Au (40 nm) at a rate of 0.5 and 1.5 Ås$^{-1}$. The lift-off was performed with acetone at room temperature. One important feature of the device design was the terminals width of 1/10th or less of the channel width. This comes from a straightforward analysis of the van der Pauw expression for charge diffusive backgrounds, where if the width of the contact is on the same order of the channel width it can lead to edge-to-edge signals ~20 times in magnitude larger than the center-to-center signal.

**Electrical characterization.** The devices were characterized in a Quantum Design physical property measurement system using standard four-probe direct current measurement methods. The measurements were performed using a Keithley 2182A as current source and a Keithley 6221 as voltmeter. The gate voltage was applied using one channel of the Keithley 2636. In all measurements, the excitation current was 10 μA. Before measuring the devices, we performed an in situ annealing at 400 K for 3 h, with helium flushing cycles to release the sample chamber of evaporated water. The samples would then be cooled down at the maximum rate to 2 K, and any temperature-dependent study performed for increasing temperatures.

**Tight-binding simulation.** The system is described by a standard nearest-neighbor tight-binding model with the magnetic field incorporated using the Peierl's phase approach $H = \sum_{\langle i,j \rangle} t_0 \exp\left(\frac{2\pi i e}{h} \int_{r_i}^{r_j} A(r) \cdot dr\right) \hat{c}_i^\dagger \hat{c}_j$, where $B = \nabla \times A$. The transmissions between each set of probes is given by the Caroli formula $T_{pq} = Tr\left[G^R \Gamma_q G^A \Gamma_p\right]$ with the required Green's functions ($G^R$, $G^A$) calculated using efficient recursive techniques. The left and right leads are included via broadening terms ($\Gamma$) calculated from the surface Green's function of semi-infinite nanoribbons, whereas constant broadening terms, representing metallic contacts, are chosen for the top and bottom probes. The Zeeman term is included separately via equal and opposite energy shifts of the spinless transmissions by half the required splitting (10$^{-3}$ eV). The total transmission is then the sum of the two spin contributions (our model excludes spin-mixing terms). The nonlocal resistance is calculated from the potentials $V_p$ and currents $I_p$ at each probe, which emerge from solving the multi-terminal Landauer–Büttiker relation $I_p = \frac{2e}{h} \sum_q \left( T_{qp} V_p - T_{pq} V_q \right)$ with suitable boundary conditions. Additional potential terms are included to represent the presence of charge inhomogeneities (electron–hole puddles) in the system induced by the graphene/substrate interaction, and the random nature of experimental GBs is accounted for by random local potentials in the vicinity of the GB. Additional details and simulations are presented in Supplementary Note 5.

**Data availability.** All relevant data are available from the authors.

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

## Acknowledgements

This work was supported by the European Union Seventh Framework Programme under the Marie Curie Actions (607904-13-SPINOGRAPH), by the Spanish MINECO (Project No. MAT2015-65159-R) and by the Regional Council of Gipuzkoa (Project No. 100/16). CIC nanoGUNE is supported by the Spanish MINECO under the Maria de Maeztu Units of Excellence Programme (MDM-2016-0618). S.R. acknowledges funding from Spanish MINECO and the European Regional Development Fund (Project No. FIS2015-67767-P (MINECO/FEDER)), and the Catalan Government (Secretaria d'Universitats i Investigació del Departament d'Economia i Coneixement). S.R.P. and S.R. acknowledge funding from the European Union Horizon 2020 Programme under the Marie Sklodowska-Curie grant agreement No. 665919 and No. 696656 Graphene Flagship. ICN2 is supported by the Spanish MINECO under the Severo Ochoa Centres of Excellence Programme (SEV-2013-0295) and by the Catalan Government under the CERCA Programme.

## Author contributions

M.R. and F.C. designed the experiments. M.R. fabricated the samples, performed the measurements, and did the data analysis. S.R.P. did the tight-binding simulations. M.R. wrote the paper. All authors participated in the discussion of the results and contributed to the writing of the manuscript. L.E.H. and F.C. supervised the entire study.
