## [Peer Review File · Nature Communications]

Reviewers' comments:

Reviewer #1 (Remarks to the Author):

The manuscript by M. Ribeiro et al., reports the non-local voltage generation in the CVD grown graphene samples under the application of perpendicular magnetic field. Their device configuration is similar to the previous non-local experiments in the exfoliated graphene that investigate spin Hall effect. However, the observation of a clear signal which is an order of magnitude larger than the Ohmic contribution in the macroscale devices indicates that the origin is not due to pure spin current. Authors discuss that this non-local signal is due to Zeeman-split counter propagating edge states, originated from the CVD graphene's grain boundaries. I think the results merits publication in Nature Communications. However, some additional control experiments need to be done, as detailed below. These comments must be addressed before publication:

1-) Did authors duplicate the result shown in Figure 5b with different devices?

2-) Authors should provide the data for R_{nl} vs V_{bg} under the external fields (similar to Figure 5b) by systematically controlling the density of line defects within the junction. A comparison study with (under a geometry similar to the modelling configuration shown in Fig S4-a) and without line defects in two separate junctions would support the main claim of the manuscript.

3-) Authors should repeat the measurement shown in Figure 5b for other aspect ratios as well for the device having 5 μm width. This is important for the completeness of the study.

4-) First spin hall effect and spin transport in large scale CVD were demonstrated in doi:10.1038/ncomms5748 and 10.1021/nl200714q, respectively. Valley hall effect in graphene was also studied by M.Yang et al., Physica E, 88, 182-187 (2017). I recommend authors to cite these works as well.

Reviewer #2 (Remarks to the Author):

The manuscript describes a non-locality measured in CVD graphene independent of the device's size, which has not been studied in CVD materials nor at that scale before. The authors describe detailed the background and their methods.

I have to say, the manuscript is extremely well written, clearly formulated in language and content! The approach is an honor for science skipping the obvious conclusion (NL SHE) and methodically investigating the phenomena behind. The conclusion is well formulated and plausible. This result is extremely interesting for other researchers in the field of spintronics and graphene electronics, where these effects can massively influence the interpretation of their measurements. I wish more researchers would demonstrate such a thorough and thoughtful analysis in their papers. I do not see any problems publishing this manuscript in the current state!

Best regards and good luck!
André Dankert

Reviewer #3 (Remarks to the Author):

The authors present an interesting analysis of how line defects in CVD graphene can lead to a new type of Bperp-dependent nonlocal signal, which might otherwise be interpreted as arising from spin, valley, or thermal currents. The proposed mechanism is based on spin-resolved counter-propagating edge modes, which travel along line defects in the graphene.

The experiment and analysis is interesting, and a valuable contribution to the nonlocal-effects-in-

graphene community, but not appropriate for Nature Communications. It is an example of how macroscale defects in graphene break the assumption of homogeneous transport that is required for ohmic behavior, and how a particular kind of these defects behave at the onset of the quantum Hall regime, but it is a paper that will be primarily of interest to a specialized audience.

With that in mind, let me make a suggestion for clarification that the authors may want to consider before resubmitting elsewhere. I find the authors' demonstration of micro-to-macroscale similar effects to be very convincing, and clear evidence that the observed effects have nothing to do with non-conserved quantities like spin, valley, or thermal currents. However, the authors did not spend enough time (in my view) explaining their proposed model.

This explanation begins on line 250, and proceeds to motivate clearly the presence of spin polarized counter propagating edge states as well as the need for bulk conduction to explain the observed B_{\perp} asymmetry. At that point, however, they rush through what seems to me to be the crux of the argument, that electron-hole asymmetry of line defects is ultimately responsible for the asymmetry. How? I would have expected that opposite B at charge neutrality simply reverses which of the edge corresponds to spin-up and spin-down, which to electrons and which to holes. How does this lead to the observed B asymmetry?

The authors try to back up their argument with a tight-binding calculation, but not nearly enough details are presented to let the reader understand the simulations (I am referring here to the description in the methods section and what is in the supplement). How, for example, electron-hole asymmetry is introduced? How large is the system in the simulation? How does size effect the observations? What characteristics of Fig 5a are random, based on the random location of the line defect, and which are generic? Why are two peaks vs one peak clearly seen in the two field orientations of the experiment, whereas just one highly disordered (sometimes split) peak is seen in the theory? Are the two clearly separated peaks for $B = -6T$ in the experiment supposed to correspond to the split "peak" in the theory?

Honestly, the simulations and the data do not look that much like each other, and the clear and simple asymmetry in the data is absent from the simulations. That is an unusual situation! Similarly, why is $65T$ needed for the simulation whereas the effect can clearly be seen at $6T$ in the experiment? That seems artificial, as $65T$ generates a Zeeman splitting large enough to create significant spin-polarization of edges, whereas $6T$ in the presence of disorder seems like more of a perturbation.

More or less my objection is this: the most interesting new aspect of this paper is the story of how counter-propagating edges along line defects can give the observed nonlocal signals. The fact that nonlocal signals can arise in many different ways, and are hard to interpret, is hardly new, especially in the quantum Hall regime. But the counter-propagating edges story is given very little serious justification, and most of that is relegated to the supplement! I hope that in the final version of this paper (probably for another journal) this part of the story will be significantly strengthened.

ANSWER TO REVIEWER #1

The manuscript by M. Ribeiro et al., reports the non-local voltage generation in the CVD grown graphene samples under the application of perpendicular magnetic field. Their device configuration is similar to the previous non-local experiments in the exfoliated graphene that investigate spin Hall effect. However, the observation of a clear signal which is an order of magnitude larger than the Ohmic contribution in the macroscale devices indicates that the origin is not due to pure spin current. Authors discuss that this non-local signal is due to Zeeman-split counter propagating edge states, originated from the CVD graphene's grain boundaries. I think the results merits publication in Nature Communications.

REPLY: We thank the reviewer for his/her appreciation of our work and hope to provide further experimental evidences and discussions to strengthen the manuscript.

However, some additional control experiments need to be done, as detailed below. These comments must be addressed before publication:

1-) Did authors duplicate the result shown in Figure 5b with different devices?

R1: Yes. The strong asymmetry of the nonlocal signals with the sign of the magnetic field was not exclusive to the micrometer sized sample, although they were more evident. This is already visible in figures 2b and 3b of the manuscript where, by sweeping the magnetic field, we see that the nonlocal signals at the charge neutrality point (CNP) develop more strongly for one of the field directions, both for the 5- μm -wide sample and for the 500- μm -wide one. In the direction of the referee's question, our reply to question number 3 (**R3**) reinforces the consistency of our results by showing that for all aspect ratios of the micrometer scale device the nonlocal resistance depends strongly on the sign of the external magnetic field. These data are now shown in the new Supplementary Figure 4. Additionally, we included a panel in the new Supp. Fig. 4 with the strong asymmetry also visible in the same fashion for the 500- μm -wide sample. As expected, for the larger devices of the 500- μm -wide sample, the averaged effect of the individual grain boundaries leads to an incomplete suppression of the CNP. We included some text in the manuscript to indicate the presence of this content.

2-) Authors should provide the data for R_{nl} vs V_{bg} under the external fields (similar to Figure 5b) by systematically controlling the density of line defects within the junction. A comparison study with (under a geometry similar to the modelling configuration shown in Fig S4-a) and without line defects in two separate junctions would support the main claim of the manuscript.

R2: Although in principle it would be possible only for the smallest scale devices, the realization of such controlled study goes far beyond the current available technical capabilities. The problem lies with the random growth of grain boundaries (GBs) in CVD graphene and the identification of all GBs or charge inhomogeneities shorting two edges within the device under study. In other works where the role of GBs has been studied in graphene [Lahiri, J., *Nat. Nanotechnol.* **5**, 326–329 (2010), Ref. 29 in the main body of the manuscript], they relied on point-probe techniques to locally identify and characterize the effects of single GBs surrounding pristine regions of graphene at the nanometer scale. In our case, with devices micrometer long and wide, we do not find it likely to successfully perform such systematically controlled experiments. On the other hand, our study comprises a wide range of device dimensions far greater than the size of the GBs, in which the effects of individual GBs have to be averaged out

and in principle should lead to smaller asymmetries of the nonlocality with the sign of B , and smaller nonlocal signals. This is exactly what we find out and report in our range of samples.

3-) *Authors should repeat the measurement shown in Figure 5b for other aspect ratios as well for the device having 5 μm width. This is important for the completeness of the study.*

R3: As indicated in the reply to the first question (**R1**), we agree that this is an important point and we have included in the new Supplementary Figure 4 the measurements of the nonlocal resistance versus gate voltage for the different aspect ratios, for external magnetic fields of -6 T and 6 T, for the 5- μm -wide sample. We found the data to be consistent with the measurements in Fig. 5b, confirming the presence for all aspect ratios of the asymmetric behavior of the nonlocality with the applied external magnetic field. This further corroborates our analysis, indicating the same mechanism to be strongly affecting the nonlocal response for all aspect ratios of the microscale device. We have included a paragraph in the main manuscript discussing this data and its availability in the new Supplementary Note 4.

4-) *First spin hall effect and spin transport in large scale CVD were demonstrated in doi:10.1038/ncomms5748 and 10.1021/nl200714q, respectively. Valley hall effect in graphene was also studied by M.Yang et al., Physica E, 88, 182-187 (2017). I recommend authors to cite these works as well.*

R4: We thank you for the suggestions and we have included the proposed references in the manuscript. Please note that reference doi:10.1038/ncomms5748 was already in the manuscript (Ref. 8), while the two other references were added. Reference doi:10.1021/nl200714q is now Ref. 7 and M. Yang *et al.*, Physica E **88**, 182-187 (2017) is now Ref. 19.

ANSWER TO REVIEWER #2

The manuscript describes a non-locality measured in CVD graphene independent of the device's size, which has not been studied in CVD materials nor at that scale before. The authors describe detailed the background and their methods.

I have to say, the manuscript is extremely well written, clearly formulated in language and content! The approach is an honor for science skipping the obvious conclusion (NL SHE) and methodically investigating the phenomena behind. The conclusion is well formulated and plausible. This result is extremely interesting for other researchers in the field of spintronics and graphene electronics, where these effects can massively influence the interpretation of their measurements. I wish more researchers would demonstrate such a thorough and thoughtful analysis in their papers. I do not see any problems publishing this manuscript in the current state!

REPLY: We would like to thank Reviewer #2 for recommending our work for publication and for being so enthusiastic about its relevance and of the methodology employed to reach our conclusions.

ANSWER TO REVIEWER #3

The authors present an interesting analysis of how line defects in CVD graphene can lead to a new type of B_{perp} -dependent nonlocal signal, which might otherwise be interpreted as arising from spin, valley, or thermal currents. The proposed mechanism is based on spin-resolved counter-propagating edge modes, which travel along line defects in the graphene.

The experiment and analysis is interesting, and a valuable contribution to the nonlocal-effects-in-graphene community, but not appropriate for Nature Communications. It is an example of how macroscale defects in graphene break the assumption of homogeneous transport that is required for ohmic behavior, and how a particular kind of these defects behave at the onset of the quantum Hall regime, but it is a paper that will be primarily of interest to a specialized audience.

With that in mind, let me make a suggestion for clarification that the authors may want to consider before resubmitting elsewhere. I find the authors' demonstration of micro-to-macroscale similar effects to be very convincing, and clear evidence that the observed effects have nothing to do with non-conserved quantities like spin, valley, or thermal currents. However, the authors did not spend enough time (in my view) explaining their proposed model.

R1: We thank the referee for his/her comments and careful reading of our work. Given the prominent role that nonlocal measurements now play in many high-impact scenarios (including detecting signatures of spin and valley Hall effects), we feel that our work has a broader audience than the referee suggests. This is particularly relevant given that the trends in nonlocal resistances are commonly used to infer on exotic phenomena in 2D materials. Ultimately, this is due to the strong desire of the scientific community to release spintronics from the cumbersome magnetic elements in the device architectures and to rely solely on the non-magnetic transporting channel (graphene) to perform all the tasks of generating, transporting and detecting spin (or valley) currents. Thus, an exact understanding of how such quantities vary in CVD graphene, the most likely candidate for large-scale production of graphene devices, is vital. It is also essential to understand the mechanisms leading to the decay or suppression of such signals, as this behaviour is commonly interpreted in terms of spin or valley relaxation times.

However, we agree with the referee that the mechanism driving the B -field asymmetry may not have been explained in enough detail, and this is now addressed in this re-submission. We have significantly extended the discussion in the supplementary information on the origin of the magnetic field asymmetry in nonlocal resistances, and updated some elements in the manuscript to clarify this issue.

This explanation begins on line 250, and proceeds to motivate clearly the presence of spin polarized counter propagating edge states as well as the need for bulk conduction to explain the observed B_{perp} asymmetry. At that point, however, they rush through what seems to me to be the crux of the argument, that electron-hole asymmetry of line defects is ultimately responsible for the asymmetry. How? I would have expected that opposite B at charge neutrality simply reverses which of the edge corresponds to spin-up and spin-down, which to electrons and which to holes. How does this lead to the observed B asymmetry?

R2: The author is correct that an opposite B reverses the spin polarisation of edge states, but its effect on an electron-hole asymmetric channel is more complex. To address the referee's

concerns, an additional supplementary section, Supplementary Note 5, and Supplementary Figures 5, 6, and 7, have been included, demonstrating explicitly how an electron-hole asymmetry is the key ingredient for the B -field asymmetry in R_{NL} at the charge neutrality point. The text in the body of the manuscript has also been adjusted to reflect this analysis.

*The authors try to back up their argument with a tight-binding calculation, but not nearly enough details are presented to let the reader understand the simulations (I am referring here to the description in the methods section and what is in the supplement).
How, for example, electron-hole asymmetry is introduced?*

R3: We agree that our simulations were not presented in enough detail, and have rectified this in a rewritten Supplementary Note 6, showing a clearer illustration of both the methods and the points we wish to demonstrate.

As stated in the manuscript, the electron-hole asymmetry for pristine grain boundaries is related to the local breaking of sublattice symmetry – the electron-hole symmetry generally associated with graphene emerges naturally from the equivalence of the two sublattices in the honeycomb structure, which is broken when a grain boundary is introduced. In experimental systems, electron-hole symmetry can also come from local potential fluctuations due to substrate effects. Our simulations include both effects, and the supplementary material dealing with tight-binding simulations has been extended and rewritten to clarify this point.

*How large is the system in the simulation?
Hoes does size effect the observations?*

R4: The system is 40 nm wide and 85 nm long, as now stated explicitly in the manuscript. A calculation at full experimental scale is unwieldy, and not necessary to produce the main feature – a nonlocal signal which is highly asymmetric with magnetic field.

What characteristics of Fig 5a are random, based on the random location of the line defect, and which are generic?

R5: As discussed in the new Supplementary Notes 5 and 6, the position of the grain boundary relative to at least one probe is important in finding the asymmetry in systems with low backscattering and almost ballistic edge channels. The grain boundary position is less important in systems with backscattering or bulk-mediated scattering. This is now demonstrated explicitly in Supplementary Note 5 using a Landauer-Buttiker analysis. A strong suppression of the nonlocal resistance is generic to almost all GB types, with the strength of the suppression and its asymmetry depending on individual GB characteristics. We have added more examples of this behaviour in the expanded supplementary information.

*Why are two peaks vs one peak clearly seen in the two field orientations of the experiment, whereas just one highly disordered (sometimes split) peak is seen in the theory?
Are the two clearly separated peaks for $B=-6T$ in the experiment supposed to correspond to the split “peak” in the theory?*

R6: The split peaks in the experimental curve are associated with the 1st Landau level (LL1). We note that the nonlocal mechanism discussed in this work is based on an electron-hole argument only valid near the CNP. Away from this point we expect R_{NL} to follow ρ_{xx} , which has peaks at the Landau levels [see Abanin *et al.*, Science **332**, 328-330 (2011); Gorbachev *et al.*,

Science **346**, 448-451 (2014)]. The secondary theoretical peaks in our original manuscript did not correspond to these, as those simulations focused only on energies near the CNP. To avoid this confusion we now show the theoretical simulation over a larger energy range, where we note similar features to the experimental curves at the expected LL1 positions. Figure 5a of the manuscript has been updated to reflect the discussion.

Honestly, the simulations and the data do not look that much like each other, and the clear and simple asymmetry in the data is absent from the simulations. That is an unusual situation!

R7: We would argue that the main qualitative feature, namely the large asymmetry in the nonlocal resistance at the CNP, is present in both systems. However, we believe that the qualitative match, although present in our initial manuscript, is far more convincing in the updated and rescaled simulations presented in this version (new Fig. 5a). Furthermore, the extended Supplementary Information now shows the range of behaviours that can be expected for different systems. Importantly, please note that the B -field asymmetry is not explained by the alternative mechanisms discussed in the literature, but our mechanism describes a considerable number of observations that have been so far unaddressed in the works exploring spin and valley Hall effects using nonlocal Hall bar geometries.

Similarly, why is 65T needed for the simulation whereas the effect can clearly be seen at 6T in the experiment? That seems artificial, as 65T generates a Zeeman splitting large enough to create significant spin-polarization of edges, whereas 6T in the presence of disorder seems like more of a perturbation.

R8: We are sorry that our original explanation in the supplementary information was somewhat confusing on this point. The Zeeman and orbital (Hall) effects of the external magnetic field are treated separately in our simulations.

The effect of the Zeeman term in our model is to split the up and down bands, and define the width (in energy) of the regime where dissipative Hall / field-induced QSHE is dominant. This value is set at $\Delta Z \sim 10^{-3}$ eV, which is in the expected range for experimental systems with $B \sim 10$ T when, e.g., interaction effects are included [see Abanin *et al.*, Phys. Rev. B **79**, 35304 (2009) and Abanin *et al.*, Phys. Rev. Lett. **107**, 96601 (2011)].

The orbital effects, included via the Peierls' phases, induce the quantum Hall effect and its resultant edge states. They also determine the energy scale of the Landau levels. Our simulations are limited to significantly smaller devices than in experiment, and require a smaller cyclotron radius to ensure that the QH edge states are properly confined. This in turn requires a larger value of B .

We note the key ingredients for a non-local response are robust against these approximations:

- The electron-hole asymmetry through a grain boundary is present over a wide range of B
- The appearance of a nonlocal peak occurs for any value of ΔZ .

These effects are now explained clearly in the Supplementary Information.

More or less my objection is this: the most interesting new aspect of this paper is the story of how counter-propagating edges along line defects can give the observed nonlocal signals. The fact that nonlocal signals can arise in many different ways, and are hard to interpret, is hardly new, especially in the quantum Hall regime. But the counter-propagating edges story is given

very little serious justification, and most of that is relegated to the supplement! I hope that in the final version of this paper (probably for another journal) this part of the story will be significantly strengthened.

R9: We believe that the additional arguments we have made, both experimentally by demonstrating the persistence of a qualitative asymmetry with B -field, and theoretically by the additional analysis and calculation, strengthen our arguments significantly.

LIST OF CHANGES

- Four references added: Ref. 5, Ref. 7, Ref. 19, and Ref. 26 of the manuscript. 1 Supplementary Reference added.
- New text has been introduced along the paper, which we highlighted in yellow in the revised manuscript.
- 2 new Supplementary Notes added (Note 4 and 5), and 1 Supplementary Note revised (Note 6). 5 new Supplementary Figures added as well (Supp. Figs. 4, 5, 6, 7, and 8). The supplementary information has been greatly updated, with the new added elements highlighted in yellow.
- Figure 5a of the manuscript has been updated to include a wider energy range in the simulation, and Fig 5b has been rescaled.

Reviewers' comments:

Reviewer #1 (Remarks to the Author):

I have read the responses to reviewers' comments. While they have not fully address my point #2 which i believe it to be very critical in order to directly present the line defects origin of the signal, I am overall satisfied with the improvement of the paper. I recommend for publication.

Reviewer #3 (Remarks to the Author):

The authors have somewhat improved their paper based on referee suggestions, but my primary judgement remains the same, that this is not appropriate for Nature Communications.

1. The authors argue that this paper belongs in a high profile journal in order to highlight realistic limitations against the dream of making spin or valley electronics in graphene. That argument might have held some weight 5 years ago, but the illusion of graphene as an "easy" material to use for spin- or valley-tronics has been broken by many papers, by many authors, in recent years. This is particularly true for CVD graphene, for which there is really no expectation (within the community) of "perfect" graphene behaviours over macroscopic distances.

Therefore, I maintain my impression that this paper belongs in a more specialized journal as an example of yet another way that nonlocal signals can arise in systems that combine disorder, inhomogeneity, and quantum Hall effect.

2. The most interesting aspect of this paper is the particular mechanism of B asymmetry identified by the authors, which they believe is at play in this experiment. However, I still do not believe the authors do a very good job of convincing the reader that this mechanism is the right one to explain the data. Most of the details are in the supplement, which is an appropriate location to store away experimental and theoretical details but not appropriate to put the main figures and arguments on which the most interesting story in the paper is built. When I look carefully at the supplement, I find that arguments presented by the authors are even LESS believable than when I read only the main text. This makes it particularly deceptive to relegate all the mechanism details to the supplement, where most readers will never see them!

Basically, I do not believe that a macroscopic sample, averaging over thousands or millions of grain boundaries with random e-h asymmetry, can give a nonlocal signal that is dramatically clearer and cleaner than a tight binding calculation covering a single grain boundary in a nanometer-scale sample, see Fig 5 a vs b and many more examples in the supplement. The experimental signature is extremely clear: large nonlocal signals at the CNP for +B and absolutely nothing for -B, with LL1 peaks for -B and absolutely nothing for +B. The theoretical curves are most more random, indeed showing some asymmetry but not at all resembling the all-or-nothing characteristic of the data, especially after one would average over macroscopic numbers of randomly asymmetric GBs.

One last question, related to this issue of averaging over randomly asymmetric GBs: why do the authors suppose that the peak in RNL at the CNP occurs in 6 out of 7 datasets for +B, with only one for -B? [F5b and SF4a-e vs SF4f] And, the one time that the peak occurs for -B it does not have the all-or-none character of many of the other datasets? Perhaps the authors have identified a signature of e-h asymmetry that is not random, due to defects at GBs or a particular puddle distribution, but due to water absorption or the substrate?

REPLY TO REFEREE #3

Reviewer #3 (Remarks to the Author):

1. The authors argue that this paper belongs in a high profile journal in order to highlight realistic limitations against the dream of making spin or valley electronics in graphene. That argument might have held some weight 5 years ago, but the illusion of graphene as an "easy" material to use for spin- or valley-tronics has been broken by many papers, by many authors, in recent years. This is particularly true for CVD graphene, for which there is really no expectation (within the community) of "perfect" graphene behaviours over macroscopic distances. Therefore, I maintain my impression that this paper belongs in a more specialized journal as an example of yet another way that nonlocal signals can arise in systems that combine disorder, inhomogeneity, and quantum Hall effect.

ANSWER: With this comment, the referee implies that the physics of spin or valley-driven transport phenomena, initially predicted/observed in "*perfect graphene*", is irrelevant in more complex (polycrystalline) and large-scale CVD graphene. The scientific consensus of the 2D materials community totally refutes this argument. Polycrystalline 2D materials are the only foreseeable platform towards any real-world electronic application, and is a strong driving force in current research in 2D materials [K. Kang *et al.*, '*Layer-by-layer assembly of two-dimensional materials into wafer-scale heterostructures*', *Nature* (2017), doi:10.1038/nature23905]. It has become more than clear that CVD graphene is a strong platform to study spin transport and proximity effects, as evidenced by many recent publications in high-profile journals [to cite a few: M. Venkata *et al.*, '*Long distance spin communication in chemical vapor deposited graphene*', *Nat. Comm.* **6**, 6766 (2015); J. Balakrishnan, '*Giant spin Hall effect in graphene grown by chemical vapor deposition*', *Nat. Comm.* **5**, 4748 (2014); A. Dankert & Saroj P. Dash, '*Electrical gate control of spin current in van der Waals heterostructures at room temperature*' *Nat. Comm.* **8**, 16093 (2017); L. Banszerus *et al.*, '*Ballistic Transport Exceeding 28 μm in CVD Grown Graphene*', *Nano Lett.* **16** (2), 1387-1391 (2016)].

Contrary to the statements of the referee, the polycrystalline nature of large scale CVD graphene is a key concern of the community at large – the robustness of various transport phenomena driven by spin or valley degrees of freedom is of major interest from both fundamental scientific perspectives but also as a route towards the incorporation of such phenomena into devices with practical applications, see for instance the recent work by K. Kang *et al.*, *Nature*, (2017), doi:10.1038/nature23905.

2. The most interesting aspect of this paper is the particular mechanism of B asymmetry identified by the authors, which they believe is at play in this experiment. However, I still do not believe the authors do a very good job of convincing the reader that this mechanism is the right one to explain the data. Most of the details are in the supplement, which is an appropriate location to store away experimental and theoretical details but not appropriate to put the main figures and arguments on which the most interesting story in the paper is built. When I look carefully at the supplement, I find that arguments presented by the authors are even LESS believable than when I read only the main text. This makes it particularly

deceptive to relegate all the mechanism details to the supplement, where most readers will never see them!

ANSWER: This interpretation is a very limited view of our work, and we believe it can only stem from misreading our manuscript. The paper is clearly written such that the experimental results and theoretical discussion slowly lead the reader to the following conclusions: 1) that we observe non-local signals in a range of systems whose dimensions vary drastically in scale, which is completely unexpected; 2) that there is an inconsistency of such signals with an interpretation based on bulk spin- or valley-currents, which has been the recurrent interpretation in the literature so far, maybe due to the temptation to do so when the devices are at the smallest scales; 3) that these signals are consistent with a mechanism driven by counter-propagating, magnetic-field induced channels; and, finally, 4) that the unusual B-field asymmetries in the experimental results are consistent with the presence of local, electron-hole asymmetric channels through the bulk of the device, as demonstrated by our model. Each of these results is essential to the total message and interpretation of our findings. In the experimental section we demonstrate point 1, and in the discussion section we carefully present the arguments supporting points 2, 3, and 4, with reference to, as requested, explicit technical details (i.e. formal proofs, limiting cases, and additional examples) in the Supplementary Material.

We do not believe that our approach is in any way deceptive, and regard the accusation as unfair and borderline insulting. In the manuscript, we start by explaining the magnetic field asymmetry in the main text by discussing the appearance of B -field induced counter-propagating states and associated non-local resistances in the context of the existing literature. We then discuss the electron-hole symmetries that occur in line defects due to either intrinsic (sublattice) or extrinsic (doping) features. Finally, we note that electron-hole and $+B/-B$ asymmetries are coupled at the CNP by the Zeeman term. We believe that our explanation here enables most readers to quickly identify the connection between our proposed mechanism and the B -field asymmetry noted in the experimental curves. References are made to Supplementary Material sections 5 and 6, where the individual steps are explained in detail using Landauer-Buttiker channel analysis and atomistic simulations of microscale systems to demonstrate the proof-of-principle.

The Supplementary Material complements the arguments in the main manuscript, providing evidence for some of the less intuitive properties of counter-propagating edge states and non-local resistances. We are confused by the author's statement "***arguments presented by the authors are even LESS believable than when I read only the main text***". The Supplementary Material consists of clear, direct, and explicit answers to specific, technical questions raised by the same referee in the first round of review. Indeed, we note that the significant expansion of the Supplementary Material was inspired by his/her first round of comments "***...I would have expected that opposite B at charge neutrality simply reverses which of the edges corresponds to spin-up and spin-down....How does this lead to the observed B asymmetry?***". In the Supplementary Material, we explicitly demonstrate the emergence of B-field asymmetry from an electron-hole asymmetry to satisfy similarly curious readers. These details are very technical and unnecessary in such an explicit form in the main manuscript.

3. Basically, I do not believe that a macroscopic sample, averaging over thousands or millions of grain boundaries with random e-h asymmetry, can give a nonlocal signal that is dramatically clearer and cleaner than a tight binding calculation covering a single grain boundary in a nanometer-scale sample, see Fig 5 a vs b and many more examples in the supplement. The experimental signature is extremely clear: large nonlocal signals at the CNP for +B and absolutely nothing for -B, with LL1 peaks for -B and absolutely nothing for +B. The theoretical curves are most more random, indeed showing some asymmetry but not at all resembling the all-or-nothing characteristic of the data, especially after one would average over macroscopic numbers of randomly asymmetric GBs.

ANSWER: Again, this comment must stem from misreading our paper. The “all-or-nothing” experimental signal discussed by the referee (Fig. 5b) is for the smallest experimental sample, containing only a handful of grain boundaries. We clearly show it in Fig. 4b, and even highlight some of the visible grain boundaries. We state which experimental scale we are considering when Fig. 5 is discussed in the main text. Furthermore, the dramatic nature of the asymmetry is indeed less clear in larger samples: this is evident in Fig. 2b and Fig. 3b, and, as discussed in the main text, is due to averaging over multiple grain boundaries. The theoretical curves are far from random, and the case displayed in the main text shows a significant peak only for one sign of the magnetic field. This comparison clearly highlights that the qualitative experimental features indeed emerge from our proposed mechanism. There are, of course, quantitative differences between the theoretical and experimental curves, and these stem from the exact approximations and approximations used in our calculation. The transmission of the line defect channel (which is controlled in our simulations by Anderson disorder) and the strength of the local electron-hole asymmetry (electron-hole puddles in the simulation) affect respectively the suppression and B -field asymmetry of the non-local peak. We can also argue that quantum interference effects are more prominent in the simulation due to the smaller system sizes that are achievable. These details do not affect the qualitative feature we wish to demonstrate, and as a consequence are not discussed prominently in the main text. Even so, and in order to clarify potential curious readers, in the Supplementary Material section 8 we discuss such features in detail, and demonstrate the effects of slightly different parameterizations. Again, such discussion is not necessary to interpret our key findings, but may be useful for more technical readers wishing to reproduce our findings or investigate similar systems.

4. One last question, related to this issue of averaging over randomly asymmetric GBs: why do the authors suppose that the peak in RNL at the CNP occurs in 6 out of 7 datasets for +B, with only one for -B? [F5b and SF4a-e vs SF4f] And, the one time that the peak occurs for -B it does not have the all-or-none character of many of the other datasets? Perhaps the authors have identified a signature of e-h asymmetry that is not random, due to defects at GBs or a particular puddle distribution, but due to water absorption or the substrate?

ANSWER: The referee is again misreading what is very clearly stated in the article. First, the datasets of Fig. 5b and Supp. Fig. 4e are the same, but with different range of gate voltage (Supp. Fig. 4e was included for completeness of Supp. Fig. 4). Second, Supp. Figs. 4a to 4e correspond to the devices of the 5- μ m-wide (microscopic) sample, where the R_{NL} peak at the CNP occurs for + B , while the Supp. Fig. 4f corresponds to one of the devices of the 500- μ m-wide (macroscopic) sample, where the R_{NL} peak at the CNP occurs for - B . This correspondence is clearly stated in the

figure caption. Third, the “all-or-none character” noted by the referee is only present in the smallest experimental (5- μm -wide) sample, as we explicitly state in the manuscript. In this case, the asymmetry is likely mediated by a single, or very few, grain boundary(s), not millions! As discussed above, this is entirely consistent with our proposed mechanism, and with the optical image of the 5- μm -wide sample (Fig. 4b). The $-B$ peak, not displaying this “all-or-nothing” character is from a much larger (500- μm -wide) sample, where the edge states are most likely shunted by a number of possible defect channels through the bulk, so that some degree of averaging has occurred. This is consistent with the B -field sweeps shown in Figs. 2b and 3b – the asymmetry is far more pronounced for the smaller system. This point is clearly explained towards the end of the manuscript, where we state: “*...larger samples contain more grains across their width and the asymmetric effects of individual GBs will tend to be averaged out. This will lead to smaller asymmetries but also smaller nonlocal signals due to a greater number of bulk conduction channels.*”. Fourth, in the Methods section, we explicitly refer to the in-situ annealing before any measurement to ensure that any related water content has been removed.

Reviewers' comments:

Reviewer #3 (Remarks to the Author):

The authors, who were apparently insulted by my previous reviews [I'm sorry, the reviews were not intended to be personal attacks or to impugn the motives of the authors, but were frank assessments of the paper as a reader], have made a much clearer version. Even without the yellow highlighting, the paper overall is much more clearly written.

I still find the proposed mechanism doubtful, and not sufficiently supported by modelling in the paper, but this is not the place to argue about whether the mechanism is right or not. Perhaps the authors are correct. I would just point out that the authors' assertion (line 276) about asymmetry in B requiring an explanation beyond "conventional" quantum Hall physics is not so obvious. Many conventional quantum Hall experiments, in well-behaved semiconductor 2DEGs, show strongly asymmetric-in-B ρ_{xx} characteristics--even by orders of magnitude. Typically the explanations for this asymmetry come down to conducting paths in the bulk, or edge modes bypassing contacts, but do not require graphene-specific models.

The experimental side of the paper is clear, and presents an interesting result that rules out most existing explanations for nonlocality in graphene. Whether the experimental results, together with a possible model to explain them, are appropriate for NComm is up to the editors.

ANSWER TO REVIEWER #3

The authors, who were apparently insulted by my previous reviews [I'm sorry, the reviews were not intended to be personal attacks or to impugn the motives of the authors, but were frank assessments of the paper as a reader], have made a much clearer version. Even without the yellow highlighting, the paper overall is much more clearly written.

I still find the proposed mechanism doubtful, and not sufficiently supported by modelling in the paper, but this is not the place to argue about whether the mechanism is right or not. Perhaps the authors are correct. I would just point out that the authors' assertion (line 276) about asymmetry in B requiring an explanation beyond "conventional" quantum Hall physics is not so obvious. Many conventional quantum Hall experiments, in well-behaved semiconductor 2DEGs, show strongly asymmetric-in- B ρ_{xx} characteristics--even by orders of magnitude. Typically the explanations for this asymmetry come down to conducting paths in the bulk, or edge modes bypassing contacts, but do not require graphene-specific models.

The experimental side of the paper is clear, and presents an interesting result that rules out most existing explanations for nonlocality in graphene. Whether the experimental results, together with a possible model to explain them, are appropriate for NComm is up to the editors.

We thank reviewer #3 for revising his/her criticism in light of the expanded discussion brought with the appeal letter. Following his/her last comments, we have included in the discussion section of the manuscript an additional paragraph to point out the existence of similar asymmetry features in the longitudinal resistance of mesoscopic two-dimensional electron gases, which could share some commonalities with graphene systems. We also included a comment of the evidence of the influence of edge transport in the nonlocal signals in mesoscopic quantum wires. However, we have also explicitly stressed that the origin of nonlocal resistance asymmetries reported here are unique to graphene given that they are strongly related to the physics of the zeroth Landau level (LL0), for which the coexistence of electrons and holes (and contribution of electron-hole puddles effects) has no equivalent in other types of low-dimensional conductors. Please note that we already had an implicit discussion in the manuscript on this subject when we analyzed the results of the modelling of Fig. 5a: "*Furthermore, our simulations capture additional peaks associated with the first Landau level (LL1). A finite R_{NL} is here associated with conduction through bulk states and not to counter-propagating states, which only occur near the CNP. Higher LL peaks in R_{NL} follow ρ_{xx} , a feature previously demonstrated in other works*".